# Electronic nature of charge density wave and electron-phonon coupling in kagome superconductor KV$_3$Sb$_5$

Hailan Luo [1,2,6], Qiang Gao[1,6], Hongxiong Liu[1,2,6], Yuhao Gu[1,6], Dingsong Wu[1,2], Changjiang Yi[1], Junjie Jia[1,2], Shilong Wu[1], Xiangyu Luo[1,2], Yu Xu[1], Lin Zhao[1], Qingyan Wang [1], Hanqing Mao[1], Guodong Liu[1,2], Zhihai Zhu[1], Youguo Shi[1,7✉], Kun Jiang[1,7✉], Jiangping Hu [1,2], Zuyan Xu[3] & X. J. Zhou [1,2,4,5,7✉]

The Kagome superconductors AV$_3$Sb$_5$ (A = K, Rb, Cs) have received enormous attention due to their nontrivial topological electronic structure, anomalous physical properties and superconductivity. Unconventional charge density wave (CDW) has been detected in AV$_3$Sb$_5$. High-precision electronic structure determination is essential to understand its origin. Here we unveil electronic nature of the CDW phase in our high-resolution angle-resolved photoemission measurements on KV$_3$Sb$_5$. We have observed CDW-induced Fermi surface reconstruction and the associated band folding. The CDW-induced band splitting and the associated gap opening have been revealed at the boundary of the pristine and reconstructed Brillouin zones. The Fermi surface- and momentum-dependent CDW gap is measured and the strongly anisotropic CDW gap is observed for all the V-derived Fermi surface. In particular, we have observed signatures of the electron-phonon coupling in KV$_3$Sb$_5$. These results provide key insights in understanding the nature of the CDW state and its interplay with superconductivity in AV$_3$Sb$_5$ superconductors.

[1] Beijing National Laboratory for Condensed Matter Physics, Institute of Physics, Chinese Academy of Sciences, 100190 Beijing, China. [2] University of Chinese Academy of Sciences, 100049 Beijing, China. [3] Technical Institute of Physics and Chemistry, Chinese Academy of Sciences, 100190 Beijing, China. [4] Songshan Lake Materials Laboratory, Dongguan 523808, China. [5] Beijing Academy of Quantum Information Sciences, 100193 Beijing, China. [6] These authors contributed equally: Hailan Luo, Qiang Gao, Hongxiong Liu, Yuhao Gu. [7] These authors jointly supervised this work: Youguo Shi, Kun Jiang, X. J. Zhou. ✉email: ygshi@iphy.ac.cn; jiangkun@iphy.ac.cn; xjzhou@iphy.ac.cn

The newly discovered Kagome superconductors $AV_3Sb_5$ (A = K, Rb, Cs) have attracted much attention because they provide an ideal platform to investigate the interplay of topology, electron correlation effects, and superconductivity[1,2]. In the crystal structure of $AV_3Sb_5$ (Fig. 1a), the vanadium atoms form a Kagome lattice that is a two-dimensional network of corner-sharing triangles. The metallic Kagome lattice presents a unique electronic structure characterized by a Dirac cone at the Brillouin zone corner, von Hove singularities (VHS) at the zone boundary, and a flat band throughout the entire Brillouin zone[3,4]. Such a Kagome lattice is expected to harbor topological states[3,5], fractional charges[4,6], density wave orders[3,7,8], and unconventional superconductivity[8–11]. For example, $AV_3Sb_5$ family exhibit anomolous Hall effect[12,13], although there is neither local-moment nor long-range magnetic ordering present in them[1,12,14]; unconventional charge density wave (CDW) has been revealed in $AV_3Sb_5$[15–17]. At present, the pairing symmetry of the $AV_3Sb_5$ superconductors has been extensively studied and it is still being debated whether the superconductivity is unconventional[18–22].

The family of Kagome compounds $AV_3Sb_5$ (A = K, Rb, Cs) exhibit a CDW transition at 78–103 K observed by transport measurements[1,2,15–17,23,24]. Such a CDW transition corresponds to a three-dimensional $2 \times 2 \times 2$ lattice reconstruction[15,17,19] and promotes a structural distortion with three different V–V bond lengths named as Tri-Hexagonal (TrH) structure (Fig. 1b)[15,25]. The CDW state shows an unusual magnetic response[15] that is intimately related to the anomalous Hall effect[13] and competes with superconductivity under pressure[26–31]. Understanding the electronic structure of the CDW state is essential to reveal its nature and relation to the topological state and superconductivity[25,32–36]. However, little is known about the impact of the CDW state on the electronic structure in $AV_3Sb_5$[37–39].

In this work, we carried out high-resolution angle-resolved photoemission (ARPES) measurements to investigate the nature of the CDW instability in $KV_3Sb_5$. Clear evidence of electronic structure reconstruction induced by the $2 \times 2$ CDW transition is revealed by the observation of the band and Fermi surface foldings. The band splitting and CDW gap opening on multiple bands are observed at the boundaries of both the original and $2 \times 2$ reconstructed Brillouin zones. We have clearly resolved all the Fermi surface that enables us to map out the Fermi surface- and momentum-dependent CDW gap. The signature of electron–phonon coupling has been found on the V-derived bands. These results provide key insight in understanding the origin of the CDW and its role on the exotic physical properties and superconductivity in Kagome superconductors.

## Results

**The measured and calculated Fermi surface.** High-quality $KV_3Sb_5$ single crystals are prepared by a two-steps self-flux method[1] and characterized by X-ray diffraction (see "Methods" and Supplementary Fig. 1a). The transport and magnetic measurements show that our samples exhibit a CDW phase transition at $T_{CDW} \sim 80$ K, consistent with the previous reports[1,23]. In the normal state above $T_{CDW}$, $KV_3Sb_5$ crystallizes in a hexagonal structure with the $P6/mmm$ space group, hosting a typical Kagome structure composed of vanadium Kagome net (Fig. 1a). In the CDW phase, a distortion of the V-Kagome lattice engenders the $2 \times 2$ reconstruction and forms a tri-hexagonal (TrH) structure on the V-Kagome plane (Fig. 1b)[15,25]. Such a lattice distortion leads to Brillouin zone reconstruction in the reciprocal space which can be described by three wavevectors (Fig. 1d).

Figure 1e shows the Fermi surface mapping of $KV_3Sb_5$ measured at 20 K in the CDW state. Extended momentum space that includes both the first and second Brillouin zones is covered in our measurements. This is important to obtain a complete Fermi surface since the band structures of $KV_3Sb_5$ exhibit

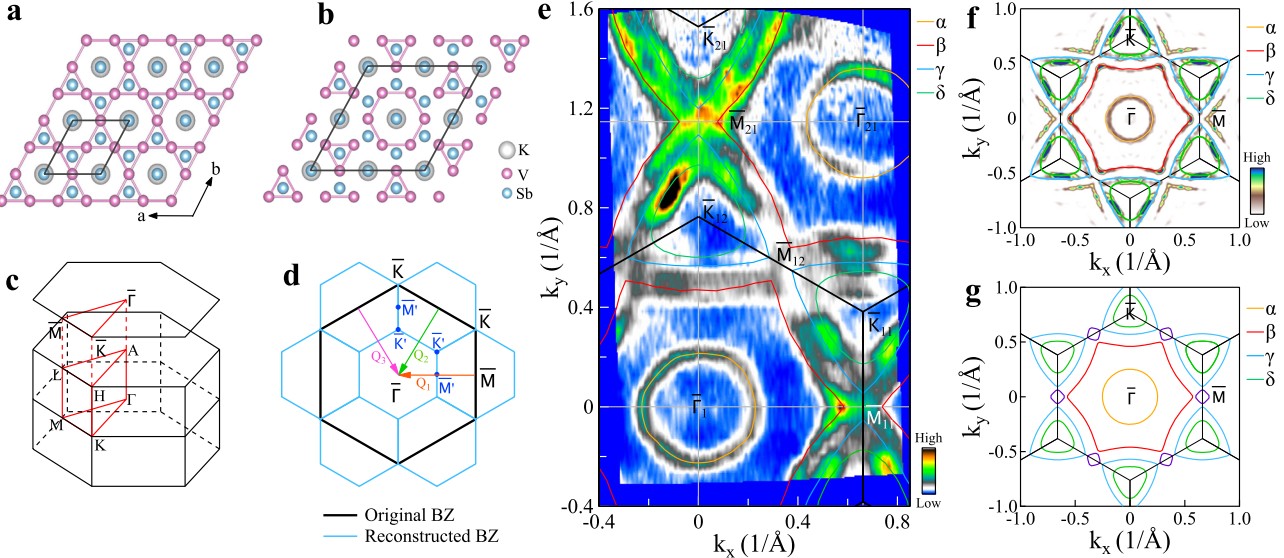

**Fig. 1 Crystal structure and Fermi surface of $KV_3Sb_5$. a** Pristine crystal structure of $KV_3Sb_5$ with a V-kagome net from the top view. **b** The Tri-Hexagonal (TrH) lattice distortion caused by the $2 \times 2$ CDW transition[15,25]. The K, V, Sb atoms are presented as gray, purple, and blue balls, respectively. **c** Schematic of the three-dimensional Brillouin zone and the two-dimensional Brillouin zone projected on the (001) surface in the pristine phase in **a**. High-symmetry points and high-symmetry momentum lines are marked. **d** The original (black lines) and $2 \times 2$ reconstructed (blue lines) Brillouin zones. The $\bar{\Gamma}$, $\bar{K}$, and $\bar{M}$ ($\bar{\Gamma}'$, $\bar{K}'$, and $\bar{M}'$) are the high-symmetry points of the pristine ($2 \times 2$ reconstructed) Brillouin zones. The arrows indicate three wavevectors (marked as $Q_1$, $Q_2$, and $Q_3$) of electronic structure reconstruction. **e** Fermi surface mapping of $KV_3Sb_5$ measured at $T = 20$ K. Four Fermi surface sheets are observed marked as $\alpha$ (orange lines), $\beta$ (red lines), $\gamma$ (blue lines), and $\delta$ (green lines). For convenience, high-symmetry points are labeled with different indexes such as $\bar{\Gamma}_1$, $\bar{\Gamma}_{21}$, $\bar{M}_{11}$, $\bar{M}_{12}$, $\bar{M}_{21}$, $\bar{K}_{11}$, $\bar{K}_{12}$, and $\bar{K}_{21}$. **f** Symmetrized Fermi surface mapping of $KV_3Sb_5$. It is obtained from taking the second derivative with respect to momentum on the second Brillouin zone data in **e**. **g** The calculated Fermi surface at $k_z = \pi/c$ corresponding to pristine crystal structure in **a**.

significant photoemission matrix element effects in different momentum space (Fig. 1e and Supplementary Figs. 2, 3). The Fermi surface mapping in Fig. 1e, combined with the analysis of the related constant energy contours (Supplementary Fig. 2) and band structures (Supplementary Fig. 3), gives rise to a Fermi surface topology that is mainly composed of a circular electron-like pocket around $\bar{\Gamma}$ ($\alpha$), a large hexagon-shaped hole-like sheet centered around $\bar{\Gamma}$ ($\beta$), a triangular hole-like pocket around $\bar{K}$ ($\gamma$) and a triangular electron-like pocket around $\bar{K}$ ($\delta$) as marked in Fig. 1e. The $\gamma$ pocket is clearly visualized around $\bar{K}_{21}$ but is weak around $\bar{K}_{12}$; its size increases with increasing binding energy in the measured constant energy contours (Supplementary Fig. 2). On the other hand, the $\delta$ pocket is clearly observed around $\bar{K}_{12}$ but is weak around $\bar{K}_{21}$; its size decreases with increasing binding energy in the constant energy contours (Supplementary Fig. 2). The quantitatively extracted Fermi surface is shown in Fig. 1f, which agree well with the calculated Fermi surface of $KV_3Sb_5$ at $k_z = \pi/c$ in its pristine structure in Fig. 1g.

**CDW-induced Fermi surface reconstruction and band folding.** The CDW-related $2 \times 2$ lattice reconstruction is expected to generate electronic structure reconstruction, as illustrated in Fig. 1d. However, no signature of such electronic reconstruction has been detected in the previous ARPES measurements[2,37–44]. We have observed clear evidence of electronic structure reconstruction induced by the $2 \times 2$ CDW transition in $KV_3Sb_5$ both in the measured Fermi surface and the band structure. Figure 2a replots the Fermi surface mapping of $KV_3Sb_5$ shown in Fig. 1e, focusing on the first Brillouin zone. In addition to the main Fermi surface, some additional features are clearly observed, as marked by the arrows in Fig. 2a. Figure 2b shows the effect of the $2 \times 2$ lattice reconstruction on the Fermi surface as induced by one of the three wavevectors, $Q_1$. The reconstructed Fermi surface sheets (dashed lines in Fig. 2b) are produced from shifting the original $\alpha$, $\beta$, $\gamma$, and $\delta$ main Fermi surface (solid lines in Fig. 2b) by the wavevector of $\pm Q_1$. As shown in Fig. 2a, the extra features can be attributed to the reconstructed Fermi surface because the observed features (1, 2), (3, 4), and 5 agree well with the reconstructed $\delta$, $\beta$, and $\alpha$ Fermi surface, respectively. Under the measurement geometry we used, the observed folded bands are mainly from $Q_1$ while those from $Q_2$ and $Q_3$ are rather weak. This is due to the photoemission matrix element effect. By changing the measurement geometry, the folded bands from other $Q_2$ or $Q_3$ wavevectors can also be observed (Supplementary Fig. 4). The electronic reconstruction is also directly evidenced in the measured band structure in Fig. 2c, in which the band measured along the $\bar{\Gamma}$-$\bar{M}$ direction coincides with the direction of $Q_1$ wavevector. As shown in Fig. 2c, in addition to the main $\beta$ bands, some extra bands are clearly observed around $\bar{\Gamma}$ ($\beta'_L$ and $\beta'_R$). The extra feature around $\bar{\Gamma}$ resembles the strong $\beta$ band at $\bar{M}$. Quantitative analysis of the momentum distribution curve (MDC) at the Fermi level in Fig. 2d indicates that the two extra features at $\bar{\Gamma}$ ($\beta'_L$ and $\beta'_R$) are separated from the $\beta$ band at $\bar{M}$ ($\beta_L$ and $\beta_R$) by exactly a wavevector of $Q_1$. Further analysis of the photoemission spectra (energy distribution curves, EDCs) at $\bar{M}$ and $\bar{\Gamma}$ in Fig. 2e indicates that they have similar lineshape near the Fermi level within an energy range of ~0.3 eV.

Figure 2f shows the Fermi surface mapping of $KV_3Sb_5$ measured by high-resolution laser-ARPES at 20 K. In addition to the main $\alpha$ Fermi surface sheet, additional features are observed which can be attributed to the reconstructed Fermi surface from the original $\beta$, $\gamma$, and $\delta$ Fermi surface sheets, as marked by the dashed guidelines. Figure 2g–i shows the temperature-dependence of the band structure measured along a momentum cut around the $\bar{\Gamma}$ point. Clear band foldings, not

only from the $\beta$ band around $\bar{M}$, but also from the $\gamma_2$ band around $\bar{M}$, can be observed at ~40 K (Fig. 2g) and ~80 K (Fig. 2h). As shown in Fig. 2a, f, the observed $\alpha$ Fermi surface, and the folded $\beta$ Fermi surface is quite similar in the measurements using 21.2 and 6.994 eV photon energies. The observed $\alpha$ bands in Fig. 2c, g are also similar although the momentum cut for the 6.994 eV measurement in Fig. 2f is slightly off the $\bar{\Gamma}$-$\bar{M}$ cut for the 21.2 eV measurement in Fig. 2a. We note that the folded band of $\gamma_{2L}$ and $\gamma_{2R}$ is prominent in Fig. 2g while it is rather weak in Fig. 2c. This difference may be attributed to the photoemission matrix element effects. These folded bands become significantly suppressed at 120 K (Fig. 2i) above the CDW temperature of ~80 K. These results strongly demonstrate that the extra features at $\bar{\Gamma}$ are replicas of the $\beta$ and $\gamma_2$ bands at $\bar{M}$ caused by the $2 \times 2$ CDW modulation.

**CDW-induced band splitting and gap opening.** Besides the electronic structure reconstruction, the manifestations of the CDW transition involve the opening of the CDW gap, both at the Fermi level and away from the Fermi level. We have clearly observed the CDW gap openings for both cases. Figure 3 shows band structures of $KV_3Sb_5$ measured along high-symmetry directions $\bar{\Gamma}$-$\bar{M}$ (Fig. 3a, d), $\bar{K}$-$\bar{M}$-$\bar{K}$ (Fig. 3b, e) and $\bar{\Gamma}$-$\bar{K}$ (Fig. 3c, f) at 20 K. For comparison, we also present the calculated band structures for both pristine (Fig. 3g) and reconstructed (Fig. 3h) crystal structures. For the reconstructed crystal structure, we calculated the effective band structure of $2 \times 2 \times 1$ TrH CDW phase of $KV_3Sb_5$ by unfolding its eigenstates in supercell Brillouin zone into primitive cell Brillouin zone (see "Calculations" section in "Methods" for more details). In the calculated band structure for the pristine lattice structure (Fig. 3g), the bands around the Fermi level originate mainly from the $5p$ orbitals of Sb ($\alpha$ band from the in-plane Sb while $\gamma_2$ band from the out-of-plane Sb) and the $3d$ orbitals of V ($\beta$, $\gamma_1$, and $\delta$ bands). The $\delta$ band originates from the V-Kagome lattice with the prototypical Dirac point at $\bar{K}$ and von Hove singularities at $\bar{M}$. The $\beta$ band also comes from V-Kagome lattice with different orbital character. The $2 \times 2$ lattice reconstruction causes significant modifications of the band structures, manifested mainly by the band splitting and the associated CDW gap opening at $\bar{M}$ in the original Brillouin zone and $\bar{M}'$ in the reconstructed Brillouin zone. As shown in the calculated band structure for the reconstructed lattice in Fig. 3h, within the energy of interest, three CDW gaps open at $\bar{M}$: $\bar{M}G1$ from $\delta_1$ band, $\bar{M}G2$ from $\zeta$ band, and $\bar{M}G3$ from $\delta_2$ band. In the meantime, four CDW gaps open at $\bar{M}'$ point: $\bar{M}PG1$ from $\delta_2$ band, $\bar{M}PG2$ from $\gamma_1$ band, $\bar{M}PG3$ from $\beta_2$ band, and $\bar{M}PG4$ from $\delta_1$ band. We note that all the bands shown in Fig. 3h are primary bands; it does not include folded bands. Therefore, all the gaps marked in Fig. 3h are real CDW gaps. In addition, the spin–orbit coupling (SOC) is expected to open a gap at the Dirac point formed from the $\delta$ bands at $\bar{K}$, as marked by DG in Fig. 3g, h.

The expected band splittings and CDW gap openings at $\bar{M}$ and $\bar{M}'$ below the Fermi level are clearly observed in the measured band structures of $KV_3Sb_5$. Figure 3d, e shows the CDW gap openings at the $\bar{M}$ point where the $\zeta$ band opens a gap labeled as $\bar{M}G2$ and the $\delta_2$ band opens a gap $\bar{M}G3$. In the corresponding EDCs at $\bar{M}$, signatures of these two gap openings can also be clearly visualized with the gap size of ~150 meV for $\bar{M}G2$ and ~125 meV for $\bar{M}G3$. Figure 3f shows the CDW gap openings at the $\bar{M}'$ point where the $\beta_2$ band opens a gap labeled as $\bar{M}PG3$ and the $\delta_1$ band opens a gap $\bar{M}PG4$. In the corresponding EDC at $\bar{M}'$ in Fig. 3j, the $\bar{M}PG4$ gap can be clearly determined with a gap size of ~150 meV. The $\bar{M}PG3$ gap is present as seen from the dip in EDC near the binding energy of 300 meV that corresponds to

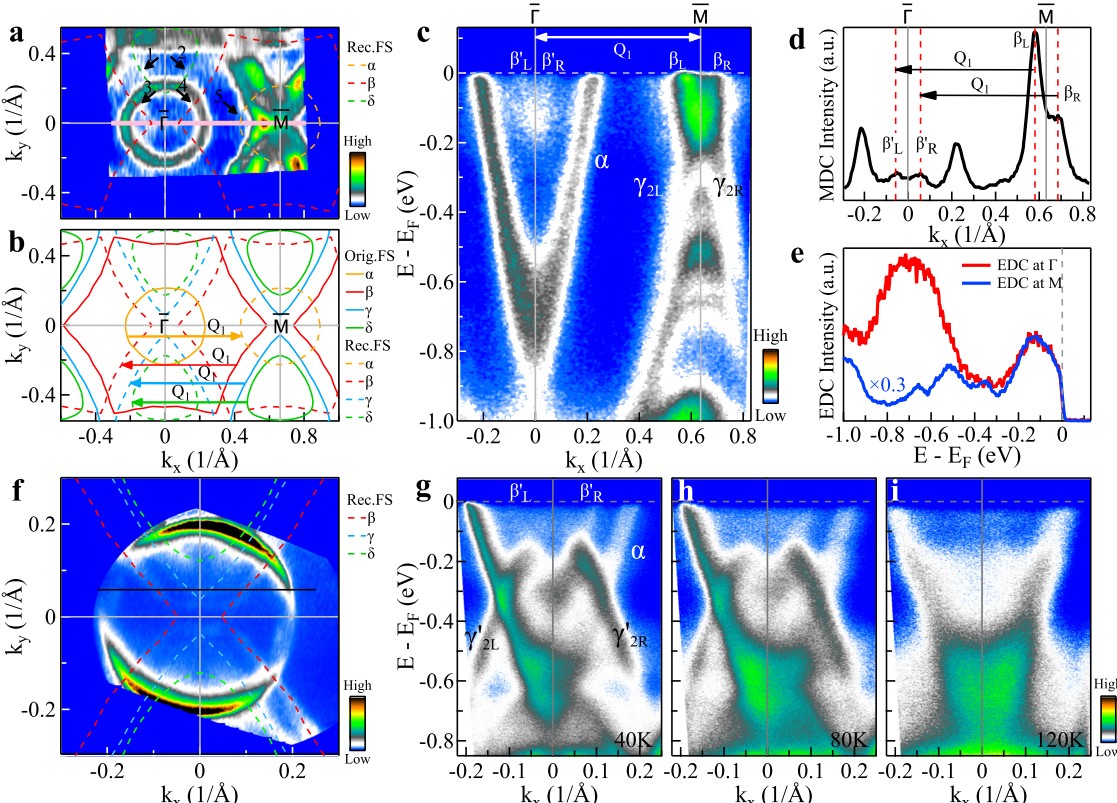

**Fig. 2 Evidence of electronic structure reconstruction in KV₃Sb₅. a** Fermi surface mapping of $KV_3Sb_5$ at 20 K in the CDW phase. In addition to the original Fermi surface, some extra weak features can be observed as marked by arrows and guided by the dashed lines that are reconstructed Fermi surface due to the CDW wavevector $Q_1$. **b** Schematic of the reconstructed Fermi surface of $KV_3Sb_5$ due to CDW wavevector $Q_1$. The solid lines represent the original Fermi surface sheets. The reconstructed Fermi surface sheets (dashed lines) are obtained by shifting the original Fermi surface with a wavevector $\pm Q_1$. **c** Band structure measured along the $\bar{\Gamma}$-$\bar{M}$ high-symmetry direction at 20 K. The location of the momentum cut is marked as a solid pink line in **a**. Around $\bar{\Gamma}$ just below the Fermi level, some extra bands can be observed. **d** The momentum distribution curve (MDC) at the Fermi level from the band structure in **c**. Two MDC peaks ($\beta_L$ and $\beta_R$) can be observed around $\bar{M}$, and another two MDC peaks ($\beta'_L$ and $\beta'_R$) can be observed around $\bar{\Gamma}$. The separation between $\beta_L$ and $\beta'_L$ ($\beta_R$ and $\beta'_R$) corresponds to the reconstruction wavevector $Q_1$. **e** The photoemission spectra (energy distribution curves, EDCs) measured at $\bar{\Gamma}$ and $\bar{M}$ points in **c**. They show similar EDC lineshape in the low binding energy region ($E_B < 0.3$ eV). **f** Fermi surface mapping of $KV_3Sb_5$ at 20 K in the CDW phase measured by laser-ARPES. Some extra weak features can be observed as marked by the dashed lines that are reconstructed Fermi surfaces due to the CDW wavevector $Q_1$. **g–i** Band structures along the momentum cut marked as a black line in **f** measured at 40 K (**g**), 80 K (**h**), and 120 K (**i**).

the spectral weight suppression in the region pointed out by the arrow in Fig. 3c. However, the related band is weak; its gap size is hard to be determined precisely but estimated to be ~150 meV. The SOC gap opening of the Dirac point at $\bar{K}$ can be seen from the EDCs in Fig. 3k; the measured gap size is ~80 meV. The measured band splittings and gap openings agree well with those from band-structure calculations.

**Fermi surface- and momentum-dependent CDW gaps.** Now we come to the CDW gap on the Fermi surface. To this end, we took high energy resolution (~4 meV) ARPES measurements on $KV_3Sb_5$ at 5 K, covering the momentum space around $\bar{M}_{21}$ as shown in Fig. 4l. In this region, in addition to the well-resolved $\alpha$ and $\beta$ Fermi surface sheets, the $\gamma$ and $\delta$ sheets are also well separated because the former is strong around $\bar{K}_{21}$ while the latter is strong around $\bar{K}_{12}$. The clearly distinguished four Fermi surface sheets facilitate the extraction of the Fermi surface-dependent and momentum-dependent CDW gaps. Figure 4a–e shows the symmetrized EDCs along the four Fermi surface; the data are taken on the two $\beta$ sheets on the two sides of $\bar{M}_{21}$ for confirming the data reliability. In the symmetrized EDCs, the gap opening causes a spectral weight suppression near the Fermi level that gives rise to a dip at the Fermi level; the gap size can be determined by the peak position near the Fermi level. The extracted CDW gaps

along the four Fermi surface sheets are plotted in Fig. 4f–j. No CDW gap opening is observed around the $\alpha$ Fermi surface as shown in Fig. 4a, f. For the $\beta$ Fermi surface, both measurements in Fig. 4b, c give a consistent result on the CDW gap in Fig. 4g, h. The CDW gap on the $\beta$ Fermi surface is anisotropic; it shows a minimum close to zero along the $\bar{\Gamma}$-$\bar{M}$ and $\bar{\Gamma}$-$\bar{K}$ directions but exhibits a maximum in the middle between these two directions. The CDW gaps along the $\gamma$ and $\delta$ Fermi surface sheets show similar behaviors, as seen in Fig. 4d, e and i, j. They are both anisotropic, showing a minimum along the $\bar{\Gamma}$-$\bar{K}$ direction and a maximum along the $\bar{K}$-$\bar{M}$ direction. The EDCs along the $\gamma$ and $\delta$ Fermi surface also show multiple features (Fig. 4d, e); besides the low energy peak, there is another peak at a higher binding energy around 70 meV. As we will show below, such a peak-dip-hump structure can be attributed to the electron–phonon coupling. Figure 4k shows a three-dimensional picture summarizing the Fermi surface-dependent and momentum-dependent CDW gaps we have observed in $KV_3Sb_5$. The CDW gaps measured along the $\beta$, $\gamma$, and $\delta$ Fermi surface sheets show strong momentum anisotropy with only small portions of the Fermi surface ungapped. We have checked on the origin of the CDW gap anisotropy in terms of the Fermi surface folding picture (Supplementary Fig. 5). The observed CDW gap anisotropy is in a qualitative agreement with the expected results from the Fermi surface folding picture.

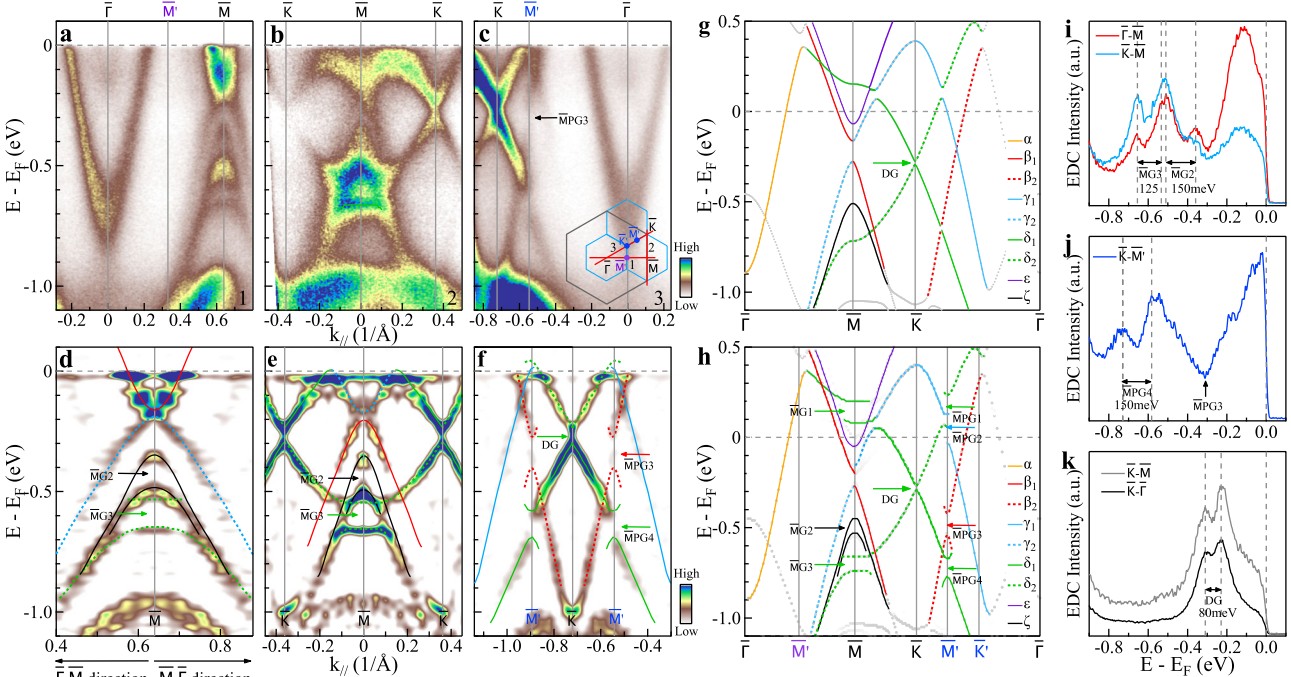

**Fig. 3 CDW-induced band splitting and gap opening in the measured band structures of KV$_3$Sb$_5$ at 20 K and their comparison with band-structure calculations. a–c** Band structures measured along the $\bar{\Gamma}$-$\bar{M}$ (**a**), $\bar{K}$-$\bar{M}$-$\bar{K}$ (**b**), and $\bar{K}$-$\bar{\Gamma}$ (**c**) high-symmetry directions, respectively. The locations of the momentum cuts, 1, 2, and 3 for (**a**), (**b**), and (**c**), respectively, are shown in the inset of **c**. **d–f** Detailed band structures around $\bar{M}$ and $\bar{K}$ points measured along $\bar{\Gamma}$-$\bar{M}$ (**d**), $\bar{K}$-$\bar{M}$-$\bar{K}$ (**e**), and $\bar{K}$- $\bar{\Gamma}$ (**f**) directions, respectively. These are symmetrized second derivative images obtained from the band structures of $k =$ 0.4–0.64 1/Å in **a**, 0 ~ 0.45 1/Å in **b**, and −0.72 ~ −0.3 1/Å in **c**, and symmetrized with respect to the $\bar{M}$ (**a**), $\bar{M}$ (**b**), and $\bar{K}$ (**c**) points, respectively. The measured band structures are indicated by guidelines and the associated CDW gaps and SOC gap are also marked. **g** Calculated band structure of KV$_3$Sb$_5$ with pristine crystal structure in Fig. 1a at $k_z = \pi/c$ with SOC. **h** The calculated band structures of KV$_3$Sb$_5$ with reconstructed TrH crystal structure in Fig. 1b at $k_z = \pi/c$ with SOC. In addition to the original high-symmetry points $\bar{\Gamma}$, $\bar{M}$ and $\bar{K}$, new high-symmetry points from the reconstructed Brillouin zone (Fig. 1c), $\bar{M}'$ and $\bar{K}'$, are marked. Three CDW gaps open at $\bar{M}$: $\bar{M}$G1, $\bar{M}$G2 and $\bar{M}$G3, and four CDW gaps open at $\bar{M}'$: $\bar{M}$PG1, $\bar{M}$PG2, $\bar{M}$PG3, and $\bar{M}$PG4. The SOC gap opening at the Dirac point at $\bar{K}$ is marked by DG. **i** EDCs at $\bar{M}$ from band structures in **a** and **b**. **j** EDC at $\bar{M}'$ from band structure in **c**. The CDW gap size is measured by the separation between related EDC peaks. **k** EDCs at $\bar{K}$ from band structures in **b** and **c**.

**Signatures of electron–phonon coupling.** The CDW transition usually involves electronic structure reconstruction and lattice distortion in which the electron–phonon coupling plays an important role[45]. We have obtained clear evidence of electron–phonon coupling in KV$_3$Sb$_5$. Figure 5a–c zooms in on the band structures of KV$_3$Sb$_5$ near the Fermi level measured along $\bar{\Gamma}$-$\bar{K}$, $\bar{K}$-$\bar{M}$-$\bar{K}$ and $\bar{\Gamma}$-$\bar{M}$-$\bar{\Gamma}$ directions at 20 K in the CDW state. The corresponding EDCs are shown in Fig. 5d–f. For the $\delta$ band in Fig. 5a, $\gamma$ and $\delta$ bands in Fig. 5b and $\beta$ band in Fig. 5c, the peak-dip-hump structure is clearly observed near their respective Fermi momenta as the peaks are marked by triangles and the humps are marked by bars in Fig. 5d–f. Figure 5g shows the expanded view of the $\delta$ band in Fig. 5a. A kink in the dispersion can be observed as marked by an arrow in Fig. 5g. The quantitative dispersion is obtained by fitting momentum distribution curves (MDCs) at different binding energies and plotted on top of the observed band in Fig. 5g. Taking a linear line as an empirical bare band, the effective real part of the electron self-energy is shown in Fig. 5h. It shows a clear peak at ~36 meV. The observed kink in the energy dispersion and the peak-dip-hump structure in EDCs are reminiscent of those from the electron–boson coupling in simple metal[46] and high-temperature superconductors[47]. The phonon frequency of the vanadium vibrations in AV$_3$Sb$_5$ can reach up to ~36 meV[32] that is consistent with the mode energy we have observed. Therefore, we have observed significant self-energy effects in KV$_3$Sb$_5$. It can be interpreted in terms of electron–phonon coupling which is present for all the $\beta$, $\gamma$ and $\delta$ bands.

## Discussion

The CDW state is first proposed for a one-dimensional chain of atoms with an equal spacing $a$ which is argued to be inherently unstable against the dimerized ground state[48]. It usually involves one band with a half electron filling. This would open a CDW gap at the Fermi point $k_F = \pm\pi/2a$ and produce a lattice reconstruction with a wavevector of $\pi/a$. Such a Fermi surface nesting picture is extended to real materials with higher dimensions where the CDW state is realized because segments of the Fermi surface are nearly parallel connected by a wavevector $Q_{CDW}$[45]. This would give rise to a partial CDW gap opening on the Fermi surface and reconstructions of both the electronic structure and the lattice with a wavevector of $Q_{CDW}$. In KV$_3$Sb$_5$, multiple Fermi surface sheets are observed with the estimated filling of 0.17 electrons/unit for the $\alpha$ pocket, 1.10 holes/unit for the $\beta$ pocket, 0.48 holes/unit for the $\gamma$ pocket, and 0.27 electrons/unit for the $\delta$ pocket (Fig. 1f). Judging from the measured Fermi surface topology, the probable nesting vectors, if exist, are along the $\bar{\Gamma}$-$\bar{K}$ direction where parts of the $\beta$, $\gamma$, and $\delta$ Fermi surface sheets exhibit nearly parallel Fermi surface. However, these vectors are not consistent with the band folding CDW wavevectors $Q_1$, $Q_2$, and $Q_3$ along $\bar{\Gamma}$-$\bar{M}$ direction directly determined by STM and X-ray diffraction measurements[15–17]. Therefore, whether the classical Peierls instability picture can be applied to the CDW formation in KV$_3$Sb$_5$ needs further investigations. Besides the Fermi surface nesting, the CDW phase can also be driven by the concerted action of electronic and ionic subsystems where a **q**-dependent electron–phonon coupling plays an indispensable

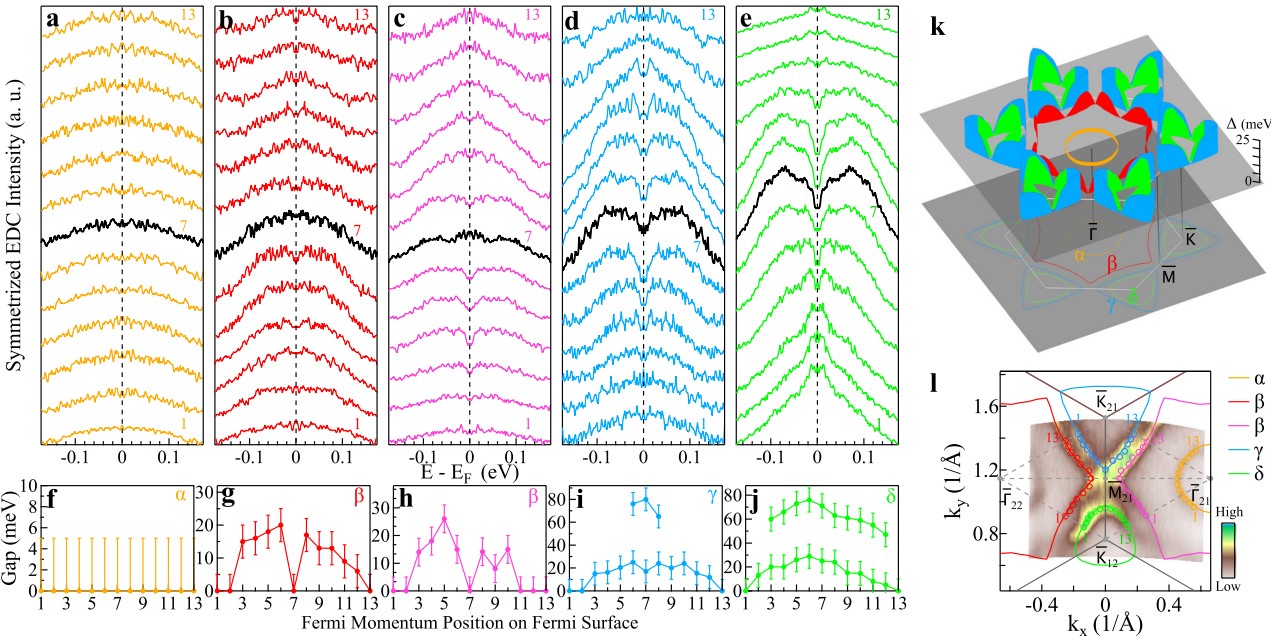

**Fig. 4 Fermi surface- and momentum-dependent CDW gaps of KV₃Sb₅ measured at 5 K. a–e** Symmetrized EDCs along the Fermi surface sheets α (**a**), β (**b, c**), γ (**d**), and δ (**e**). The corresponding Fermi momentum positions are marked in **l** by numbers on each Fermi surface sheet. **f–j** CDW gap size as a function of momentum on the Fermi surface α (**f**), β (**g, h**), γ (**i**) and δ (**j**). The gap size is obtained by picking the peak positions in the symmetrized EDCs in **a–e**. When multiple peaks are observed in **i** and **j**, the position of the higher binding energy peak is also extracted. Error bars reflect the uncertainty in determining the CDW gaps. **k** Three-dimensional plot of the Fermi surface-dependent and momentum-dependent CDW gaps in KV₃Sb₅. **l** High-resolution Fermi surface mapping of KV₃Sb₅ at 5 K. The observed four Fermi surface sheets α, β, γ, and δ are marked.

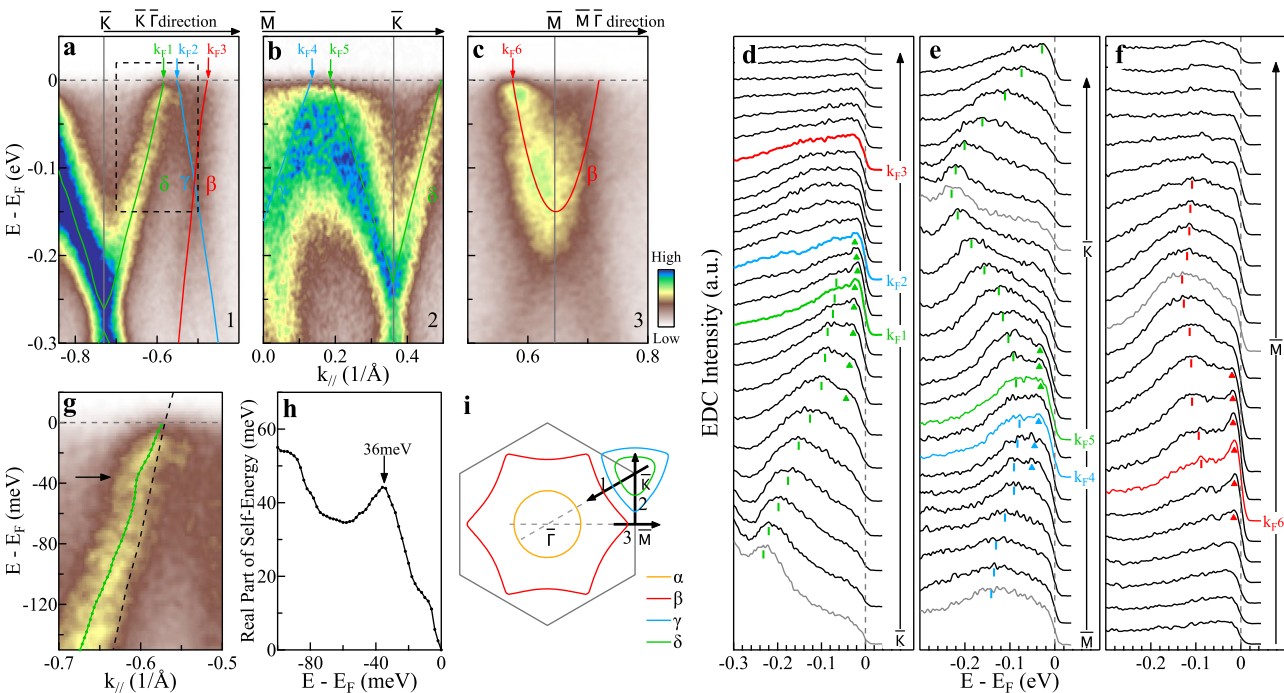

**Fig. 5 Electron–phonon coupling in KV₃Sb₅. a–c** Detailed band structures along $\bar{K}$-$\bar{\Gamma}$ (**a**), $\bar{M}$-$\bar{K}$ (**b**), and $\bar{\Gamma}$-$\bar{M}$-$\bar{\Gamma}$ (**c**) directions, respectively, measured at 20 K. The analysis of these band structures is shown in Supplementary Fig. 6. The locations of the momentum cuts, 1, 2, and 3 for **a**, **b**, and **c**, respectively, are shown in **i**. The Fermi momenta of the β, γ, and δ bands are marked by arrows and labeled by kF1 ~ kF6. **d–f** The corresponding EDCs for the band structures in **a–c**, respectively. Peak-dip-hump structure can be observed near kF1, kF2, kF4, kF5, and kF6. The EDC peaks are marked by triangles while the humps are marked by bars. **g** Expanded view of the δ band inside the dashed box in **a**. The MDC fitted dispersion is shown by the green line and the dashed black line represents the calculated band from Fig. 3h as the bare band. **h** Real part of the electron self-energy extracted from **g**. It shows a peak at ~36 meV. **i** Schematic of the Fermi surface and the locations of the momentum cuts 1, 2, and 3 for the band structures in **a**, **b**, and **c**, respectively.

part[49,50]. In AV$_3$Sb$_5$ system, the driving force for the CDW formation remains under debate[15,17,25,32,33,37,38]. Based on our observations, we found that the electron–phonon coupling plays a major role in generating the CDW phase in KV$_3$Sb$_5$. Firstly, the measured Fermi surface (Fig. 1f) and band structures (Fig. 3a–c) of KV$_3$Sb$_5$ show a high agreement with the band-structure calculations that do not incorporate the electron-electron interactions, which indicates the electron correlation effect is weak in KV$_3$Sb$_5$. Note that in the comparison, there is no Fermi level adjustment in the calculated electronic structures. Specifically, the calculated $\delta$ band bottom at $\bar{K}$ in Fig. 3h lies at the binding energy of 270 meV while the measured value of the $\delta$ band bottom at $\bar{K}$ in Fig. 5a is ~256 meV, indicating a weak band renormalization due to electron correlation. Secondly, besides the gap opening around the Fermi surface, we have also observed a clear CDW gap opening at $\bar{M}$ and $\overline{M'}$ with a gap size up to ~150 meV (Fig. 3) highly away from the Fermi level. Thirdly, the electron–phonon couplings on the $\beta$, $\gamma$, and $\delta$ bands are directly observed (Fig. 5). All these results indicate that the CDW phase in KV$_3$Sb$_5$ is mainly driven by the electron–phonon coupling induced structural phase transition.

In summary, through our high-resolution ARPES measurements and the density functional theory (DFT) calculations on KV$_3$Sb$_5$, clear evidence of the $2 \times 2$ CDW-induced electronic structure reconstruction has been uncovered. These include the Fermi surface reconstruction, the associated band-structure foldings, and the CDW gap openings at the boundary of the pristine and reconstructed Brillouin zone. The Fermi surface-dependent and momentum-dependent CDW gap is measured and strong anisotropy of the CDW gap is observed for all the V-derived Fermi surface sheets. The electron–phonon couplings have been observed for all the V-derived bands. These results indicate that the electron correlation effect in KV$_3$Sb$_5$ is weak and the electron–phonon coupling plays a dominant role in driving the CDW transition. They provide key information in understanding the origin of the CDW state and its interplay with superconductivity in AV$_3$Sb$_5$ superconductors.

## Methods

**Growth and characterization of single crystals**. High-quality single crystals of KV$_3$Sb$_5$ were grown from a two-steps flux method[1]. First, KSb$_2$ alloy was sintered at 573 K for 20 hours in an alumina crucible coated with aluminum foil. Second, high-purity K, V, Sb, and KSb$_2$ precursor were mixed in a molar ratio of 1:3:14:10 and then sealed in a Ta tube. The tube was sealed in an evacuated quartz ampoule, heated up to 1273 K, soaked for 20 h, and then cooled down to 773 K at a rate of 2 K/h. Shiny lamellar crystals were separated from the flux by centrifuging with a regular hexagon shape and a size up to $4 \times 4$ mm$^2$ (inset of Supplementary Fig. 1a). The crystals were characterized by X-ray diffraction (Supplementary Fig. 1a) and their magnetic susceptibility and resistance were measured (Supplementary Fig. 1b, c). The CDW transition temperature, $T_{CDW}$, is ~80K from the magnetic measurement in Supplementary Fig. 1b.

**High-resolution ARPES measurements**. High-resolution angle-resolved photoemission measurements were carried out on our lab system equipped with a Scienta R4000 electron energy analyzer[51,52]. We use a helium discharge lamp as the light source that can provide a photon energy of $h\nu = 21.218$ eV (helium I). The energy resolution was set at ~ 20 meV for the Fermi-surface mapping (Fig. 1) and band-structure (Figs. 2a, c, 3 and 5) measurements and at 4 meV for the CDW gap measurements (Fig. 4). We also use an ultraviolet laser as the light source that can provide a photon energy of $h\nu = 6.994$ eV with a bandwidth of 0.26 meV. The energy resolution was set at ~2.5 meV for the measurements in Fig. 2f–i. The angular resolution is ~0.3°. The Fermi level is referenced by measuring on a clean polycrystalline gold that is electrically connected to the sample. The sample was cleaved in situ and measured in a vacuum with a base pressure better than $5 \times 10^{-11}$ Torr.

**Calculations**. First-principles calculations are performed by using the Projected Augmented Wave Method (PAW) within the spin-polarized density functional theory (DFT), as implemented in the Vienna Ab Initio Simulation Package (VASP)[53–55]. We construct $2 \times 2 \times 1$ supercell to describe the TrH CDW phase of KV$_3$Sb$_5$. The crystal structures are relaxed by using the Perdew–Burke–Ernzerhof (PBE) functional[56] and zero damping DFT-D3 van der Walls correction[57] until the

forces are <0.001 eV/Å. The cutoff energy of plane-wave basis is set as 600 eV and the energy convergence criterion is set as $10^{-7}$ eV. The corresponding Brillouin zones are sampled by using a $16 \times 16 \times 10$ (for primitive cell) and a $8 \times 8 \times 10$ (for supercell) Gamma centered **k**-grid. The effective band structure is calculated by the band-unfolding method[58,59] proposed by Zunger et al. with BandUP code[60,61]. The $2 \times 2 \times 1$ TrH CDW order reconstruction is an input in our band-unfolding calculation. All the DFT calculations are ab initio without any adjustable parameters except for the standard exchange-correlation functional and pseudopotentials.

## Data availability

All data are processed by using Igor Pro 8.02 software. All data needed to evaluate the conclusions in the paper are available within the article and its Supplementary Information files. All raw data generated during the current study are available from the corresponding author upon reasonable request.

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

## Acknowledgements

This work is supported by the National Natural Science Foundation of China (Grant Nos. 11888101, 11922414, 11974404, 12074411, and U2032204), the National Key Research and Development Program of China (Grant Nos. 2016YFA0300300, 2016YFA0300602, 2017YFA0302900, 2017YFA0303100, 2018YFA0305602, and 2018YFA0704200), the Strategic Priority Research Program (B) of the Chinese Academy of Sciences (Grant Nos. XDB25000000, XDB28000000, and XDB33000000), the Youth Innovation Promotion Association of CAS (Grant No. 2017013), the Research Program of Beijing Academy of Quantum Information Sciences (Grant No. Y18G06) and the K. C. Wong Education Foundation (GJTD-2018-01).

## Author contributions

X.J.Z. and H.L. proposed and designed the research. H.L. and Q.G. performed ARPES experiments. H.L., Q.G., and X.J.Z. analyze the ARPES data. H.L., C.Y., and Y.S. contributed to crystal growth. Y.G., K.J., and J.H. contributed to DFT calculations. D.W., J.J., S.W., X.L., Y.X., L.Z., Q.W., H.M., G.L., Z.Z., Z.X., and X.J.Z. contributed to the development and maintenance of the ARPES systems and related software development. H.L., K.J., and X.J.Z. wrote this paper. All authors participated in the discussion and comment on the paper.

## Competing interests

The authors declare no competing interests.

## Additional information

**Peer Review Information** *Nature Communications* thanks the anonymous, reviewer(s) for their contribution to the peer review of this work. Peer reviewer reports are available

