## [Peer Review File · Nature Communications]

REVIEWER COMMENTS

Reviewer #1 (Remarks to the Author):

The manuscript by Hailan Luo et al. is an ARPES study of the newly discovered Kagome lattice superconductor KV₃Sb₅. This work is reporting the reconstructed electronic structure inside the charge ordering phase. Band folding, Fermi surface gaps and electron-phonon mediated self-energy effects are observed and discussed. An explosive amount of work is being produced on this and related compounds. In this view, this current manuscript is timely and will attract interest from the emerging community. The field is, however, also characterized by a kind of "gold rush" resulting in manuscripts written very quickly. This fact also has an impact on this paper. Before considering publication, several points should be improved.

(1) Data collection. The manuscript is reporting data recorded inside the charge ordered state. Band folding and band gaps are being reported. To make this convincing, temperature dependent data across the ordering temperature would be needed. Is the band folding reported in Figure 2 disappearing above the charge ordering temperature? This is a fairly easy experimental check. It could in principle also be carried out also for the Fermi surface gaps (more challenging). As a minimum, the temperature dependence of the band-folding in figure 2 should be presented.

(2) Data presentation. Figure 3 is very rich and could probably be improved. The correspondence between panel a-c and d-f is not obvious. The d-f panels seem to be symmetrized but this is not mentioned in the caption. Panel d is presumably generated from data in panel a. However, the Gamma-point indication in d is not corresponding to that of panel a. These are all details, but it is somewhat important because the reader would wish to examine the band folding in the raw data. Better correspondence the panels would therefore be helpful.

(3) Band structure calculations. It is stated that "The measured band splitting and gap openings agree well with those band structure calculations." The band structure calculations are according to the method section based on DFT methodology. Usually, it is hard for DFT calculations to capture precisely electron-phonon induced charge ordering. So, the question is whether the DFT calculations actually have been adjusted such that it agrees well with the observed data rather than the other way around? The method section seems to suggest that the 2x2 charge order reconstruction is an input rather than an output of the model. If the DFT calculation is really an ab-initio calculation without any adjustable parameters, this should be stated more clearly.

(4) Self-energy effects. It is stated that "Therefore, we have observed signatures of the electron-phonon coupling in KV₃Sb₅ ..." Maybe it would be more prudent to state that significant self-energy effects have been observed and interpreted in terms of electron-phonon coupling.

(5) Discussion. In the discussion, the one-dimensional Peierls instability leading to the formation of dimers are mentioned. It is suggested that this concept can be extended to high dimensions. Indirectly, the reader get's the impression that it can apply to KV₃Sb₅. Since the Peierls instability requires specific electron filling, it would be useful to discuss the electron filling of KV₃Sb₅. Even the 2x2 reconstruction resembles that of a classical Peierls instability the filling situation in KV₃Sb₅ might be very different.

(6) There are some typo's through out the manuscript: undstanding ->understanding, strcture ->structure, sueface -> surface

Reviewer #2 (Remarks to the Author):

The manuscript by Luo et al. presents an ARPES study of KV₃Sb₅, a topological material displaying

superconductivity, a large anomalous Hall effect and unconventional charge order at low temperature. The authors show for the first time the effect of the charge density wave on the electronic structure, which is a fundamental step to understand the behaviour of this intriguing material. Specifically, they find evidence for a 2×2 reconstruction of the Fermi surface, including gaps at the Fermi level and signatures of electron-phonon coupling.

Given the growing interest in systems where correlation effects and non-trivial topology coexist, I would expect this work to attract significant interest. However, I found the interpretation of some of the ARPES data in terms of CDW gaps not correct; there are issues in the presentation of the band structure calculation and I had the impression that the link between these results and other key properties of KV_3Sb_5 , such as band topology and the emergence of unconventional superconductivity, was very vague.

A list of points that need attention is given below.

1. The authors describe all observable splittings in the ARPES data as CDW gaps. I think this is not correct. The authors should carefully distinguish between back-folding of the bands due to the 2×2 reconstruction and real gaps. Considering the band structure calculation in Fig.3 (g), it seems clear that the CDW gap labelled MG2 is not a gap. This apparent splitting is due to the back-folding of a band with a maximum at Gamma onto the M point.
2. I was puzzled by the band structure calculation displayed in Fig. 3. The authors should clearly explain in the main text what type of calculation they are plotting in panel (h). Why are there bands of different grey colour? Why is the band structure at two equivalent M' points totally different? Some of the bands highlighted in green suddenly disappear half way between different high symmetry points; why is that?
3. The authors write that the reconstructed Fermi surface can be explained by shifting the original bands by the wavevector Q1. I wonder why they they do not see the reconstructions from the other two wavevectors Q2 and Q3.
4. The CDW gaps measured along the different Fermi surface contours show strong momentum anisotropy with only very small portions of the Fermi surface ungapped. The authors should comment on the origin of this anisotropy and explain to what extent it is related to nesting of the Fermi surface.
5. The authors find kinks in the low energy dispersion which they interpret as signatures of electron phonon coupling. In order to extract the real part of the self energy they use an empirical linear bare band. It would be more appropriate to use as bare band the calculated band structure. This would also give an accurate idea on the strength of electronic correlation. Based on the overall agreement between band structure calculations and ARPES data the authors claim that correlation effects should be weak but a comparison at the quantitative level is missing.

Response to Reviewer's Comments

Reviewer #1 (Remarks to the Author):

The manuscript by Hailan Luo et al. is an ARPES study of the newly discovered Kagome lattice superconductor KV₃Sb₅. This work is reporting the reconstructed electronic structure inside the charge ordering phase. Band folding, Fermi surface gaps and electron-phonon mediated self-energy effects are observed and discussed. An explosive amount of work is being produced on this and related compounds. In this view, this current manuscript is timely and will attract interest from the emerging community. The field is, however, also characterized by a kind of “gold rush” resulting in manuscripts written very quickly. This fact also has an impact on this paper. Before considering publication, several points should be improved.

(1) Data collection. The manuscript is reporting data recorded inside the charge ordered state. Band folding and band gaps are being reported. To make this convincing, temperature dependent data across the ordering temperature would be needed. Is the band folding reported in Figure 2 disappearing above the charge ordering temperature? This is a fairly easy experimental check. It could in principle also be carried out also for the Fermi surface gaps (more challenging). As a minimum, the temperature dependence of the band-folding in figure 2 should be presented.

(2) Data presentation. Figure 3 is very rich and could probably be improved. The correspondence between panel a-c and d-f is not obvious. The d-f panels seem to be symmetrized but this is not mentioned in the caption. Panel d is presumably generated from data in panel a. However, the Gamma-point indication in d is not corresponding to that of panel a. These are all details, but it is somewhat important because the reader would wish to examine the band folding in the raw data. Better correspondence the panels would therefore be helpful.

(3) Band structure calculations. It is stated that “The measured band splitting and gap openings agree well with those band structure calculations.” The band structure calculations are according to the method section based on DFT methodology. Usually, it is hard for DFT calculations to capture precisely electron-phonon induced charge ordering. So, the question is whether the DFT calculations actually have been adjusted such that it agrees well with the observed data rather than the other way around? The method section seems to suggest that the 2x2 charge order reconstruction is an input rather than an output of the model. If the DFT calculation is really an ab-initio calculation without any adjustable parameters, this should be stated more clearly.

(4) Self-energy effects. It is stated that “Therefore, we have observed signatures of the electron-phonon coupling in KV₃Sb₅ ...” Maybe it would be more prudent to state that significant self-energy effects have been observed and interpreted in terms of electron-phonon coupling.

(5) Discussion. In the discussion, the one-dimensional Peierls instability leading to the formation of dimers are mentioned. It is suggested that this concept can be extended to high dimensions. Indirectly, the reader get's the impression that it can apply to KV3Sb5. Since the Peierls instability requires specific electron filling, it would be useful to discuss the electron filling of KV3Sb5. Even the 2x2 reconstruction resembles that of a classical Peierls instability the filling situation in KV3Sb5 might be very different.

(6) There are some typo's through out the manuscript: undestanding ->understanding, structure ->structure, sueface -> surface

Response to Reviewer #1

We thank Reviewer #1 for the careful reviewing of our paper and his/her constructive comments and suggestions to improve our paper. We also thank the reviewer for nicely capturing the importance and significance of our work.

(1). Thank the referee for the good suggestion. We have carried out temperature-dependent measurement across the $T_{CDW} \sim 80K$ by high-resolution laser-ARPES, as shown in Fig. R1. In this case, we observe the band folding of not only the beta band, but also the epsilon band from M to Gamma. We also find that the folded bands disappear above the CDW temperature.

Following the referee's suggestion, we have added the new data in Fig. 2f-i in the revised manuscript. The related figure caption and the text are also added.

Fig. R1. Band structures of KV₃Sb₅ around the Gamma point measured at 40 K (a), 80 K (b) and 120 K (c). The beta'_L, beta'_R, gamma'_{2L} and gamma'_{2R} refer to the folded bands from M to Gamma.

(2). We have improved Fig. 3 following the referee's suggestions. We have added description in the figure caption that Fig. 3d-f are symmetrized images obtained from

Fig. 3a-c. We also modified the momentum axes of Fig. 3d-f to have a better correspondence to the momentum axes in Fig. 3a-c.

(3). The $2 \times 2 \times 1$ TrH CDW order reconstruction is an input in our band-unfolding calculation. All the DFT calculations are *ab-initio* without any adjustable parameters except for the standard exchange-correlation functional and pseudopotentials. We have made the statement more clearly in the revised manuscript by adding “The $2 \times 2 \times 1$ TrH CDW order reconstruction is an input in our band-unfolding calculation. All the DFT calculations are *ab-initio* without any adjustable parameters except for the standard exchange-correlation functional and pseudopotentials.” in the Methods section.

(4). Following the referee’s suggestion, we have modified the statement of the self-energy effects from “Therefore, we have observed signatures of the electron-phonon coupling in KV_3Sb_5 for all the beta, gamma and delta bands” to “Therefore, we have observed significant self-energy effects in KV_3Sb_5 . It can be interpreted in terms of electron-phonon coupling which is present for all the beta, gamma and delta bands.”

(5). We did not imply that the extended Peierls instability picture can be applied to KV_3Sb_5 . In the one-dimensional Peierls instability picture, it usually involves one band with a half electron filling. When extended to high-dimensions, it usually involves Fermi surface nesting where parts of the Fermi surface are nearly parallel and connected by a wavevector. In KV_3Sb_5 , multiple Fermi surface sheets are observed with the estimated filling of 0.17 electrons/unit for the alpha pocket, 1.10 holes/unit for the beta pocket, 0.48 holes/unit for the gamma pocket and 0.27 electrons/unit for the delta pocket. Judging from the measured Fermi surface topology, the probable nesting vectors are along the Gamma-K direction where parts of the beta, gamma and delta Fermi surface sheets exhibit nearly parallel Fermi surface. However, these vectors are not consistent with the band folding wavevectors Q1, Q2 and Q3 along Gamma-M direction directly determined by STM and X-ray diffraction measurements. Therefore, the classical Peierls instability picture seems not applicable to the CDW formation in KV_3Sb_5 .

Following the referee’s suggestion, we have added the following discussion in the revised manuscript: “In KV_3Sb_5 , multiple Fermi surface sheets are observed with the estimated filling of 0.17 electrons/unit for the alpha pocket, 1.10 holes/unit for the beta pocket, 0.48 holes/unit for the gamma pocket and 0.27 electrons/unit for the delta pocket. Judging from the measured Fermi surface topology, the probable nesting vectors are along the Gamma-K direction where parts of the beta, gamma and delta Fermi surface sheets exhibit nearly parallel Fermi surface. However, these vectors are not consistent with the band folding wavevectors Q1, Q2 and Q3 along Gamma-M direction directly determined by STM and X-ray diffraction measurements. Therefore, whether the classical Peierls instability picture can be applied to the CDW formation in KV_3Sb_5 needs further investigations.”

(6). We thank the referee's suggestion. We have carefully checked the manuscript and corrected these typos.

Reviewer #2 (Remarks to the Author):

The manuscript by Luo et al. presents an ARPES study of KV3Sb5, a topological material displaying superconductivity, a large anomalous Hall effect and unconventional charge order at low temperature.

The authors show for the first time the effect of the charge density wave on the electronic structure, which is a fundamental step to understand the behaviour of this intriguing material. Specifically, they find evidence for a 2 x 2 reconstruction of the Fermi surface, including gaps at the Fermi level and signatures of electron-phonon coupling.

Given the growing interest in systems where correlation effects and non-trivial topology coexist, I would expect this work to attract significant interest. However, I found the interpretation of some of the ARPES data in terms of CDW gaps not correct; there are issues in the presentation of the band structure calculation and I had the impression that the link between these results and other key properties of KV3Sb5, such as band topology and the emergence of unconventional superconductivity, was very vague.

A list of points that need attention is given below.

1. The authors describe all observable splittings in the ARPES data as CDW gaps. I think this is not correct. The authors should carefully distinguish between back-folding of the bands due to the 2 x 2 reconstruction and real gaps. Considering the band structure calculation in Fig.3 (g), it seems clear that the CDW gap labelled MG2 is not a gap. This apparent splitting is due to the back-folding of a band with a maximum at Gamma onto the M point.

2. I was puzzled by the band structure calculation displayed in Fig. 3. The authors should clearly explain in the main text what type of calculation they are plotting in panel (h). Why are there bands of different grey colour? Why is the band structure at two equivalent M' points totally different? Some of the bands highlighted in green suddenly disappear half way between different high symmetry points; why is that?

3. The authors write that the reconstructed Fermi surface can be explained by shifting the original bands by the wavevector Q1. I wonder why they do not see the reconstructions from the other two wavevectors Q2 and Q3.

4. The CDW gaps measured along the different Fermi surface contours show strong momentum anisotropy with only very small portions of the Fermi surface ungapped. The authors should comment on the origin of this anisotropy and explain to what extent it is related to nesting of the Fermi surface.

5. The authors find kinks in the low energy dispersion which they interpret as signatures

of electron phonon coupling. In order to extract the real part of the self energy they use an empirical linear bare band. It would be more appropriate to use as bare band the calculated band structure. This would also give an accurate idea on the strength of electronic correlation. Based on the overall agreement between band structure calculations and ARPES data the authors claim that correlation effects should be weak but a comparison at the quantitative level is missing.

Response to Reviewer #2

We thank Reviewer #2 for the careful reviewing of our paper and his/her constructive comments and suggestions to improve our paper. We also thank the reviewer for nicely capturing the importance and significance of our work.

(1). We appreciate referee for this detailed observation. In terms of calculated CDW bands, we need to emphasize that the black bands around MG2 are not coming from the band at Gamma point, although they look similar. First, the energy of the black band top at M point is -447meV while the band top at Gamma point is -450meV (Tab. R1). Secondly, the eigenstates of these two bands are different: the Gamma point band at -450meV is mainly composed by V's d orbital while the M point band at -447meV is mainly composed by Sb's p orbital. Hence, the black bands indeed open a CDW gap at MG2.

In terms of folding picture, we have carefully checked between back-folding of the bands due to the 2 x 2 reconstruction and real gaps. All the bands shown in Fig. 3h are primary bands; it does not include folded bands. Therefore, all the gaps marked in Fig. 3h are real CDW gaps. The MG2 in Fig. 3h is a CDW gap. It is not from the back folding of the band at Gamma onto the M point. First, the black top band is a primary band from the band structure calculation; Second, this black top band has a totally different dispersion from the band at Gamma point, as shown in Fig. R2 below by shifting the band at Gamma to the M point (blue line between M-Gamma in Fig. R2), ruling out the possibility that the black top band is from the back folded band at Gamma point.

To avoid possible misunderstanding, we added in the revised manuscript the description: "We note that all the bands shown in Fig. 3h are primary bands; it does not include folded bands. Therefore, all the gaps marked in Fig. 3h are real CDW gaps."

Fig. R2. Comparison between the zeta band (top black line at M) and the folded band from Gamma to M (blue line between M and K). They exhibit totally different dispersion.

G point			M point		
k (\AA^{-1})	E- E_F (eV)	Spectral weight	k (\AA^{-1})	E- E_F (eV)	Spectral weight
0.0000	-0.4580	0.000E+00	0.6688	-0.4580	0.000E+00
0.0000	-0.4570	0.000E+00	0.6688	-0.4570	0.000E+00
0.0000	-0.4560	0.000E+00	0.6688	-0.4560	0.000E+00
0.0000	-0.4550	0.000E+00	0.6688	-0.4550	0.000E+00
0.0000	-0.4540	0.000E+00	0.6688	-0.4540	0.000E+00
0.0000	-0.4530	0.000E+00	0.6688	-0.4530	0.000E+00
0.0000	-0.4520	0.000E+00	0.6688	-0.4520	0.000E+00
0.0000	-0.4510	0.000E+00	0.6688	-0.4510	0.000E+00
0.0000	-0.4500	1.995E+00	0.6688	-0.4500	1.579E-03
0.0000	-0.4490	0.000E+00	0.6688	-0.4490	0.000E+00
0.0000	-0.4480	0.000E+00	0.6688	-0.4480	0.000E+00
0.0000	-0.4470	1.254E-03	0.6688	-0.4470	6.663E-01
0.0000	-0.4460	0.000E+00	0.6688	-0.4460	0.000E+00
0.0000	-0.4450	0.000E+00	0.6688	-0.4450	0.000E+00
0.0000	-0.4440	0.000E+00	0.6688	-0.4440	0.000E+00
0.0000	-0.4430	0.000E+00	0.6688	-0.4430	0.000E+00
0.0000	-0.4420	0.000E+00	0.6688	-0.4420	0.000E+00
0.0000	-0.4410	0.000E+00	0.6688	-0.4410	0.000E+00
0.0000	-0.4400	0.000E+00	0.6688	-0.4400	0.000E+00

Tab. R1. Calculated band structures at Gamma and M points of KV_3Sb_5 .

(2). Fig. 3h displays the calculated band structures of KV_3Sb_5 with reconstructed TrH crystal structure. We calculate the eigenstates of $2 \times 2 \times 1$ TrH CDW phase KV_3Sb_5 with VASP code and then unfold them to get the effective band structure with BandUP code. As a result, the CDW distortion causes the effective band structure different from the original band structure in KV_3Sb_5 . To further clarify, we have added related description in the revised manuscript: “For the reconstructed crystal structure, we calculated the effective band structure of $2 \times 2 \times 1$ TrH CDW phase of KV_3Sb_5 by unfolding its eigenstates in supercell Brillouin zone into primitive cell Brillouin zone (see Calculations in Methods for more details).”

The original calculated band structures are plotted in grey color with the grey scale corresponding to the spectral weight of the bands, as shown in Fig. R3 below. For the convenience to describe the bands of interests, we selected some of the bands and

plotted them with different colors, leaving a few of the bands colored in grey in Fig. 3h.

We thank the referee for pointing out the M' point notation issue. We used M' to represent the middle of the edge of the reconstructed smaller Brillouin zones (small hexagons in the inset of Fig. 3c). Because the band structures in Fig. 3h represent the primary band, the band structure at M' along the Gamma-K direction (right side of Fig. 3h) is different from the one at M' along the Gamma-M direction (left side of Fig. 3h). To avoid possible misunderstanding, in the revised manuscript, we mark these two M' with different colors in Fig. 3h and also mark their positions in the inset of Fig. 3c.

We thank the referee for pointing out this problem. We have plotted the bands on both sides of M as green in the revised Fig. 3h.

Fig. R3. The calculated band structures of KV_3Sb_5 with the reconstructed TrH crystal structure. They are plotted in grey color with the grey scale corresponding to the spectral weight of the bands.

(3). Under the measurement geometry we used, the observed folded bands are mainly from Q1 while those from Q2 and Q3 are rather weak. This is likely due to the photoemission matrix element effect. To test, we carried out two ARPES measurements by rotating the sample 60° with each other while keeping all the other measurement conditions the same. As shown in Fig. R4 below, before the sample rotation, the folded bands are mainly from Q1 wavevector (Fig. R4a). After the sample is rotated clockwise by 60° , the observed folded bands are mainly from Q2 wavevector (Fig. R4b). This measurement demonstrates that the folded bands from different CDW wavevectors can be observed under different measurement geometries.

To clarify the issue, we have added Fig. R4 into the Supplementary Materials and the related description in the revised manuscript: “Under the measurement geometry we used, the observed folded bands are mainly from Q1 while those from Q2 and Q3 are rather weak. This is due to the photoemission matrix element effect. By changing the

measurement geometry, the folded bands from other Q2 or Q3 wavevectors can also be observed, as shown in Fig. S4 in Supplementary Materials.”

Fig. R4. Constant energy contours at $E_B=150\text{meV}$ of KV_3Sb_5 measured by laser-ARPES at 20K. The sample in (b) is rotated clockwise by 60° with respect to the one in (a) while keeping all the other measurement conditions the same.

(4). We thank the referee for the suggestion. We have carefully checked on the origin of the CDW gap in terms of the Fermi surface folding picture. As shown in Fig. R5 below, we plotted the original Fermi surface and the folded Fermi surface. The CDW gap is expected to open at the crossing points between the original and folded Fermi surfaces. For the beta Fermi surface, two crossing points (1 and 2 in Fig. R5) are formed near the middle between A and X points in Fig. R5. This would give a CDW gap maximum near the middle between A and X points and gap minima at A and X points. This is consistent with our measured results in Fig. 4. For the gamma Fermi surface, the observed crossing point 3 is close to the Fermi surface tip B point while the other crossing point 4 is near the middle between B and Y points. This would give a CDW gap maximum close to B point and a gap minimum at Y point. This is also consistent with the observed CDW gap on the gamma Fermi surface in Fig. 4. For the delta Fermi surface, the two crossing points 5 and 6 are close to the Fermi surface tip C point. This would give rise to a CDW gap maximum near C point and a gap minimum at Z point. This is also consistent with the measured result in Fig. 4. Overall, the observed CDW gap anisotropy is in a qualitative agreement with the expected results from the Fermi surface folding picture.

We have added Fig. R5 into the Supplementary Materials and the discussion on the origin of the CDW gap anisotropy in the revised manuscript: “The CDW gaps measured along the beta, gamma and delta Fermi surface sheets show strong momentum anisotropy with only small portions of the Fermi surface ungapped. We have checked on the origin of the CDW gap anisotropy in terms of the Fermi surface folding picture, as shown in Fig. S5 in Supplementary Materials. The observed CDW gap anisotropy is

in a qualitative agreement with the expected results from the Fermi surface folding picture.”

Fig. R5. The expected CDW gap openings in KV_3Sb_5 . The original Fermi surface alpha, beta, gamma and delta are plotted in solid lines while the folded Fermi surface sheets are plotted in dashed lines. The crossing points between the original and folded Fermi surface sheets are marked by solid circles and numbers. The CDW gap is expected to open at these crossing points.

(5). We thank the referee’s good suggestion. In the revised manuscript, we have replaced the empirical linear bare band by the calculated band in Fig. 5g. We have also added the comparison between the measured and calculated electronic structures on the quantitative level: “Note that in the comparison, there is no Fermi level adjustment in the calculated electronic structures. Specifically, the calculated delta band bottom at K in Fig. 3h lies at the binding energy of 270 meV while the measured value of the delta band bottom at K in Fig. 5a is ~ 256 meV, indicating a weak band renormalization due to electron correlation.”

Summary of changes:

1. Following the suggestion of Reviewer #1, we add the Fermi surface and temperature-dependent band structures around Gamma measured by laser-ARPES in Fig. 2f-i. The related figure caption is also added. We add corresponding text on page 5, line 125: “Figure 2f shows the Fermi surface mapping of KV_3Sb_5 measured by high-resolution laser-ARPES at 20 K. In addition to the main alpha Fermi surface sheet, additional features are observed which can be attributed to the reconstructed Fermi surface from the original beta, gamma and delta Fermi surface sheets, as marked by the dashed guidelines. Fig. 2g-i show the temperature-dependence of the band structure measured along a momentum cut around the Gamma point. Clear band foldings, not only from the beta band around M, but also from the gamma_2 band around M, can be observed at ~ 40 K (Fig. 2g) and ~ 80 K (Fig. 2h). These folded bands become significantly suppressed at 120 K (Fig. 2i) above the CDW temperature of ~ 80 K.”
2. Following the suggestion of Reviewer #1, we change “These are second derivative images obtained from the band structures in (a-c)” into “These are symmetrized second derivative images obtained from the band structures of $k=0.4\sim 0.64$ $1/\text{\AA}$ in (a), $0\sim 0.45$ $1/\text{\AA}$ in (b) and $-0.72\sim -0.3/\text{\AA}$ in (c) and symmetrized with respect to the M (a), M (b) and K (c) points, respectively.” in the figure caption of Fig. 3.
3. Following the suggestion of Reviewer #1, we have made the statement more clearly by adding “The $2\times 2\times 1$ TrH CDW order reconstruction is an input in our band-unfolding calculation. All the DFT calculations are *ab-initio* without any adjustable parameters except for the standard exchange-correlation functional and pseudopotentials.” in the Methods section of revised manuscript.
4. Following the suggestion of Reviewer #1, on page 8, line 220, we change “Therefore, we have observed signatures of the electron-phonon coupling in KV_3Sb_5 for all the beta, gamma and delta bands.” into “Therefore, we have observed significant self-energy effects in KV_3Sb_5 . It can be interpreted in terms of electron-phonon coupling which is present for all the beta, gamma and delta bands.”
5. Following the suggestion of Reviewer #1, we add some discussion on page 8, line 231: “In KV_3Sb_5 , multiple Fermi surface sheets are observed with the estimated filling of 0.17 electrons/unit for the alpha pocket, 1.10 holes/unit for the beta pocket, 0.48 holes/unit for the gamma pocket and 0.27 electrons/unit for the delta pocket. Judging from the measured Fermi surface topology, the probable nesting vectors, if exist, are along the Gamma-K direction where parts of the beta, gamma and delta Fermi surface sheets exhibit nearly parallel Fermi surface. However, these vectors are not consistent with the band folding CDW wavevectors Q1, Q2 and Q3 along Gamma-M direction directly determined by STM and X-ray diffraction

measurements. Therefore, whether the classical Peierls instability picture can be applied to the CDW formation in KV_3Sb_5 needs further investigations.”

6. Following the comment of Reviewer #1, we corrected all typos in our revised manuscript.
7. Following the suggestion of Reviewer #2, we add some discussion on page 6, line 155: “We note that all the bands shown in Fig. 3h are primary bands; it does not include folded bands. Therefore, all the gaps marked in Fig. 3h are real CDW gaps.”
8. Following the suggestion of Reviewer #2, we add some discussion on page 5, line 141: “For the reconstructed crystal structure, we calculated the effective band structure of $2 \times 2 \times 1$ TrH CDW phase of KV_3Sb_5 by unfolding its eigenstates in supercell Brillouin zone into primitive cell Brillouin zone (see Calculations in Methods for more details).”
9. Following the suggestion of Reviewer #2, to avoid possible misunderstanding, in the revised manuscript, we mark these two M’ with different colors in Fig. 3h and also mark their positions in the inset of Fig. 3c. We also plot the bands on the both sides of M as green in the revised Fig. 3h.
10. Following the suggestion of Reviewer #2, we have added Fig. R4 into the Supplementary Materials and the related description in the revised manuscript, on page 4, line 111: “Under the measurement geometry we used, the observed folded bands are mainly from Q1 while those from Q2 and Q3 are rather weak. This is due to the photoemission matrix element effect. By changing the measurement geometry, the folded bands from other Q2 or Q3 wavevectors can also be observed, as shown in Fig. S4 in Supplementary Materials.”
11. Following the comment of Reviewer #2, we have added Fig. R5 into the Supplementary Materials and the discussion on the origin of the CDW gap anisotropy in the revised manuscript, on page 7, line 198: “The CDW gaps measured along the beta, gamma and delta Fermi surface sheets show strong momentum anisotropy with only small portions of the Fermi surface ungapped. We have checked on the origin of the CDW gap anisotropy in terms of the Fermi surface folding picture, as shown in Fig. S5 in Supplementary Materials. The observed CDW gap anisotropy is in a qualitative agreement with the expected results from the Fermi surface folding picture.”
12. Following the comment of Reviewer #2, in the revised manuscript, we have replaced the empirical linear bare band by the calculated band in Fig. 5g. We have also added the comparison between the measured and calculated electronic structures on the quantitative level, on page 9, line 247: “Note that in the comparison, there is no Fermi level adjustment in the calculated electronic

structures. Specifically, the calculated delta band bottom at K in Fig. 3h lies at the binding energy of 270 meV while the measured value of the delta band bottom at K in Fig. 5a is ~256 meV, indicating a weak band renormalization due to electron correlation.”

13. We add descriptions about our 7-eV laser-based ARPES in Methods on page 10, line 286: “We also use ultraviolet laser as the light source that can provide a photon energy of $h\nu=6.994$ eV with a bandwidth of 0.26 meV. The energy resolution was set at ~2.5 meV for the measurements in Fig. 2f-i.”

REVIEWERS' COMMENTS

Reviewer #1 (Remarks to the Author):

The authors have addressed previous comments and suggestions satisfactory. Therefore, I recommend publication of this work in Nature Communications.

Reviewer #2 (Remarks to the Author):

The authors have revised the manuscript and included new data following the comments of both referees. I am satisfied with their response and I would like to recommend the manuscript for publication in Nature Comm.

However, prior to publication, there is a (small) point that remains to be addressed.

The authors should mention and explain in the main text the difference between the two dispersion plots in Fig.2(c) and 2(d). Specifically, I wonder why the replicas of the γ_{2L} and γ_{2R} bands are so prominent in the new laser-ARPES data, while they are completely invisible in the synchrotron data. Some explanations are needed to convince the readers that the two ARPES measurements are looking at the same system.

Response to Reviewer's Comments

Reviewer #2 (Remarks to the Author):

The authors have revised the manuscript and included new data following the comments of both referees. I am satisfied with their response and I would like to recommend the manuscript for publication in Nature Comm.

However, prior to publication, there is a (small) point that remains to be addressed.

The authors should mention and explain in the main text the difference between the two dispersion plots in Fig.2(c) and 2(d). Specifically, I wonder why the replicas of the gamma2L and gamma2R bands are so prominent in the new laser-ARPES data, while they are completely invisible in the synchrotron data. Some explanations are needed to convince the readers that the two ARPES measurements are looking at the same system.

Response to Reviewer #2

We thank Reviewer #2 for reviewing our paper again and providing further comments and suggestions to improve our paper.

Following the referee's suggestion, we have added the following sentences to explain the difference between Fig. 2c and 2g in the revised manuscript: "As shown in Fig. 2a and 2f, the observed alpha Fermi surface and the folded beta Fermi surface are quite similar in the measurements using 21.2 eV and 6.994 eV photon energies. The observed alpha bands in Fig. 2c and 2g are also similar although the momentum cut for the 6.994 eV measurement in Fig. 2f is slightly off the Gamma-M cut for the 21.2 eV measurement in Fig. 2a. We note that the folded band of gamma2L and gamma2R is prominent in Fig. 2g while it is rather weak in Fig. 2c. This difference may be attributed to the photoemission matrix element effects."